# Comparison of Functional Connectivity in the Prefrontal Cortex during a Simple and an Emotional Go/No-Go Task in Female versus Male Groups: An fNIRS Study

**DOI:** 10.3390/brainsci11070909

**Published:** 2021-07-09

**Authors:** Thien Nguyen, Emma E. Condy, Soongho Park, Bruce H. Friedman, Amir Gandjbakhche

**Affiliations:** 1Eunice Kennedy Shriver National Institute of Child Health and Human Development, National Institute of Health, 49 Convent Drive, Bethesda, MD 20814, USA; thien.nguyen4@nih.gov (T.N.); emma.condy@nih.gov (E.E.C.); soongho.park@nih.gov (S.P.); 2Department of Psychology, Virginia Tech, 109 Williams Hall, Blacksburg, VA 24061, USA; bhfriedm@vt.edu

**Keywords:** behavioral inhibition, emission error, commission error, response time, inhibitory control deficits, connectivity–performance correlation

## Abstract

Inhibitory control is a cognitive process to suppress prepotent behavioral responses to stimuli. This study aimed to investigate prefrontal functional connectivity during a behavioral inhibition task and its correlation with the subject’s performance. Additionally, we identified connections that are specific to the Go/No-Go task. The experiment was performed on 42 normal, healthy adults who underwent a vanilla baseline and a simple and emotional Go/No-Go task. Cerebral hemodynamic responses were measured in the prefrontal cortex using a 16-channel near infrared spectroscopy (NIRS) device. Functional connectivity was calculated from NIRS signals and correlated to the Go/No-Go performance. Strong connectivity was found in both the tasks in the right hemisphere, inter-hemispherically, and the left medial prefrontal cortex. Better performance (fewer errors, faster response) is associated with stronger prefrontal connectivity during the simple Go/No-Go in both sexes and the emotional Go/No-Go connectivity in males. However, females express a lower emotional Go/No-Go connectivity while performing better on the task. This study reports a complete prefrontal network during a simple and emotional Go/No-Go and its correlation with the subject’s performance in females and males. The results can be applied to examine behavioral inhibitory control deficits in population with neurodevelopmental disorders.

## 1. Introduction

Inhibitory control deficits are seen across a variety of conditions, including neurodevelopmental disorders, such as attention deficit hyperactivity disorder (ADHD) [1], as well as in the aging process [2]. Due to its impact across developmental stages, understanding the neural mechanisms of behavioral inhibition in typically developing populations is a necessary step in developing assessments and interventions for this ability. Relatedly, this developmental range and the disorders affected by behavioral inhibition deficits make the use of an accessible, easy-to-tolerate (e.g., robust to movement, easy to transport) technology necessary to ensure it can be successfully utilized across these populations. Simple laboratory tasks of behavioral inhibition, such as the Go/No-Go (GNG) task [3], paired with accessible neuroimaging technologies could be immensely helpful in identifying a mechanism behind these behaviors that can be targeted therapeutically in populations with deficits in this area. However, steps must first be taken to verify this approach in populations with typical neurodevelopment to determine its utility.

The GNG task is designed to measure the motor inhibitory response [3] and has been widely used to assess inhibition in neuroimaging studies. During the task, a series of “Go” and “No-Go” stimuli are presented to a subject, who is required to respond to a “Go” stimulus, but not to a “No-Go” stimulus. Repeated presentations of the “Go” stimulus create a prepotent urge to respond during the trials, making inhibition of this prepotent response during “No-Go” stimuli challenging. Previous research using functional magnetic resonance imaging (fMRI) has shown that the GNG task evokes brain activation in the prefrontal cortex, which suggests the critical role of this brain region in controlling response inhibition [4]. However, the GNG task can take on a variety of forms, which use different stimuli in the same general paradigm to interrogate behavioral inhibition ability.

Reviews of fMRI studies investigating behavioral inhibition using the GNG task show that frontal areas such as the pre-supplementary motor area, insula, and medial prefrontal cortex are activated during these tasks [5,6]. However, it is evident that as task parameters change, the areas of activation also differ. Activation likelihood estimation (ALE) meta-analyses further investigating how the varied parameters of GNG tasks affect activation found that alterations in the task complexity or “Go” to “No-Go” trial ratio, the number of “No-Go” stimuli and working memory load (e.g., more complex stimuli, more than one “Go” or “No-Go” target) changed which areas were activated [6,7]. Specifically, the authors note that certain areas, such as the right dorsolateral prefrontal, inferior parietal circuits [6], and the pre-supplementary motor area [7], may not play a direct role in inhibition because activation in these areas appear to be attributed to increased working memory load [7]. Moreover, an ALE meta-analysis by Gavazzi et al. revealed different networks for different types of inhibitory phases with the right inferior frontal gyrus associated with proactive inhibition and the right middle frontal gyrus corresponding to reactive inhibition [8]. Additional neuroimaging studies using the GNG in both its simple form and in more complex versions of the task are warranted to better evaluate the potential differences between such forms of the task.

One variation of the simple GNG task is the emotional GNG [9]. The emotional GNG uses faces with neutral or emotional expressions as the stimuli in a GNG framework, requiring input from both the medial prefrontal cortex associated with a simple GNG task [5,6] as well as the ventral prefrontal cortex involved in emotional processing [10]. The emotional GNG task is shown to be more challenging than a nonemotional GNG, evidenced by more errors and slower responses than the nonemotional version. However, scores on the emotional GNG are correlated with nonemotional GNG performance, indicating that it still appropriately measures behavior inhibition [11]. The emotional GNG was of particular interest due to the relationship between behavior inhibition and emotion regulation, shown by differences in behavior inhibition elicited by altering emotional context [11] as well as their shared neural architecture [12]. The interplay of these processes makes the emotional GNG particularly interesting, as the use of emotional stimuli may modulate behavioral inhibition ability while adding complexity to the simple GNG paradigm. This is particularly compelling when disorders associated with behavioral inhibition deficits are examined, such as ADHD, as they also show emotion regulation challenges [13,14]. In addition, there exists evidence suggesting that emotional GNG performance may relate to estrogen variation, specifically activation in the dorsolateral prefrontal cortex while inhibiting response to positive stimuli was positively correlated with luteal phase estradiol, and it was significantly increased during the luteal (high estrogen), compared to the follicular (low estrogen) phase [15]. Furthermore, females were reported to be better at emotion recognition tasks than males [16,17]. Based on these critiques and evidence, the present study used multiple versions of the GNG task to investigate differences in connectivity between a simple and complex task, and sex was included as a covariate in the analyses to control for its potential effect on emotional GNG performance.

Additionally, the majority of studies on the neural associations of behavioral inhibition have focused on measuring areas of activation, not network-level dynamics. Cerebral functional connectivity is a measure of the temporal correlation between two separate brain regions. When there exists a statistical dependence between time series of data recorded in two different regions, these regions are considered to have functional connectivity. Previous studies have largely employed simple GNG paradigms to investigate connectivity, with evidence that greater connectivity between prefrontal areas is associated with better GNG performance [18,19]. Further, studies using fMRI and diffusion tensor imaging have demonstrated structural and functional connectivity impairments in disorders characterized by inhibitory deficits such as ADHD [20], indicating that connectivity may play a critical role in inhibition. Further exploration of these network-level dynamics in the prefrontal areas is warranted to better understand the processes underlying behavioral response inhibition across contexts.

Functional near infrared spectroscopy (fNIRS) is an optical technique that indirectly monitors brain activity through cerebral hemodynamic changes. In addition to being low cost, invulnerable to motion artifact, and highly portable in comparison to fMRI, fNIRS has an important advantage of having higher temporal resolution, which is crucial to characterizing the shape and change in the hemodynamic responses. For this reason, when a large dataset is required to obtain reliable results relating to hemodynamic activation across brain regions in the computation of the functional connectivity, fNIRS is preferred over fMRI.

Although activation of the prefrontal cortex during a GNG task has been thoroughly investigated, not many studies have focused on cerebral functional connectivity during such tasks. In this research, the prefrontal cortex is selected as a targeted region because of its association with the GNG task [5,6] and its crucial role in emotion processing [21]. The present study used fNIRS to (1) examine prefrontal connectivity during a simple and an emotional GNG task, (2) analyze the sex-based correlation between connectivity and subject’s performance, and (3) identify connections specific to a simple and an emotional GNG task in females and males. We hypothesized that both female and male groups would present a positive prefrontal connectivity–performance relation during a simple GNG task, but this positive correlation may vary depending on sex group during the emotional GNG task.

## 2. Materials and Methods

### 2.1. Participants and Experimental Protocol

The experimental protocol was approved by the National Institute of Child Health and Human Development’s Institutional Review Board (10CH0198). Parts of data from this protocol were previously published in a multimodal study examining prefrontal function in relation to measures of autonomic activity (CITE) [22]. This study included 42 healthy subjects (20 males, age 37.2 (±14.7)). Before the experiment, all subjects were required to complete a health history questionnaire and sign an informed consent letter. Subjects with history of cardiovascular disease or skin disease were excluded. During the experiment, the participant was seated comfortably in a chair and was asked to follow the instructions on a monitor in front of them.

The experimental protocol consisted of three conditions: a “vanilla” baseline [23] (6.5 min), a simple GNG task (6.5 min, Figure 1a), and an emotional GNG task (6.5 min). During the baseline, participants watched a neutral video clip (Coral Sea Dreaming: Plankton Productions and MJL Network, 2014), which helped maintain minimal engagement. After the vanilla baseline, the simple and emotional GNG tasks were displayed in a random order across participants. GNG tasks consisted of 192 trials with 144 Go and 48 No-Go trials. Each trial was 500 ms long, followed by a 1500 ± 250 ms interstimulus interval. The subject was required to press a <SPACE> bar when seeing a Go stimulus and to not press any buttons when seeing a No-Go stimulus. Letters (Y: Go; X: No-Go) were presented during the simple GNG and emotional faces (neutral: Go; happy: No-Go (24); angry: No-Go (24)) were presented during the emotional GNG task. Each subject practiced six trials before each task.

Omission error, commission error, and response time were recorded and regarded as subject’s performance. An omission error is an error committed by the participant when she/he did not press a <SPACE> bar in a Go trial. Commission errors are counted when the participant pressed the <SPACE> bar in a No-Go trial. Response time is the time interval from the letter/face that was displayed on the screen until the subject pressed the <SPACE> bar.

### 2.2. Data Recording

Cerebral hemodynamic changes were measured in the subject’s prefrontal cortex using an fNIRS device (fNIR Devices LLC, New Orleans, LA, USA). The device consists of 4 LEDs (light emitting diode) emitting near infrared light at 730 nm and 850 nm and 10 light detectors, which form 16 channels. The distance between a LED and a detector is 2.5 cm. All LEDs and detectors were embedded in a flexible head band. Before experiments, participants’ head size was measured, the forehead was cleaned, an fNIRS probe was placed on the subject’s forehead centered at Fpz, and the signal quality was examined. The projection of the 16 fNIRS channels on a brain model is shown in Figure 1b. The fNIRS signal was recorded at 2 Hz sampling rate through COBI Studio software (fNIR Devices LLC, Potomac, MD, USA).

### 2.3. Data Processing

The recorded optical intensity was converted into the hemodynamic response changes, including oxy- (HbO) and deoxy- (HbR) hemoglobin, using Beer–Lamberts’ law with a differential pathlength factor was assumed to be 6. Converted data were then bandpass filtered (0.01–0.5 Hz) and denoised. Principal component analysis (PCA) was applied to remove superficial and systemic physiological signals. Studies on resting state functional connectivity have often band-pass-filtered NIRS signals in the range of 0.01–0.1 Hz (or 0.01–0.08 Hz) to acquire the cerebral spontaneous hemodynamic change [24]. However, due to the nature of the simulation used in this study (~2 s each trial), the fNIRS signal was filtered in the range of 0.01–0.5 Hz to retain possible fast brain response to the stimulus. Our previously published work has shown that systemic physiological signals in the range of 0.1–0.5 Hz such as Mayer wave and respiratory rhythm were effectively removed by the application of PCA [25]. The preprocessed fNIRS signal was split into three datasets (baseline, simple, and emotional GNG). The Pearson correlation coefficient was then calculated from HbO data between every pair of fNIRS channels to generate a symmetric 16×16 correlation matrix per subject per condition. Finally, the correlation coefficient was converted to a z-value using Fisher transformation [24,26] to be used as functional connectivity values. A total of 120 connections were considered.

A traditional approach to identify connections that are specific to a task is to compare the task and the baseline connectivity using a statistical test (i.e., *t*-test). A connection is selected when the statistical test results in a significant difference (e.g., the task connectivity is significantly greater than the baseline connectivity). However, this approach may draw a wrong conclusion, especially in the prefrontal cortex, a part of the default mode network, which is activated when resting [27]. Here, we suggest a new method to identify task-specific connections by correlating the baseline connectivity with the GNG task connectivity. A high, positive correlation coefficient indicates that a connection with a high baseline connectivity has a high task connectivity and vice versa. In other words, that connection is activated/deactivated in both baseline and during the task (not task-specific connection). On the other hand, a low correlation coefficient indicates a disassociation between the baseline and task connectivity. As a result, a connection with a low correlation coefficient may be the connection that is specific for the task.

### 2.4. Statistical Test

To assess differences in GNG performance, a series of two-tailed, paired samples *t*-tests was conducted on functional connectivity values (*z*-values) comparing the simple GNG to the emotional GNG within each sex group, and a two-tailed, independent sample *t*-test was conducted between sex groups to compare the subject’s performance using omission errors, commission errors, and reaction time as the dependent variables. The statistical test was considered to be significant when the *p*-value was less than or equal to 0.05.

A 3-way repeated measures analysis of variance (ANOVA) test and a series of post-hoc Bonferroni tests were performed to compare the connectivity of each connection across subjects between the baseline and the tasks for each sex group. Within subject factors in the ANOVA are functional connectivity values during the baseline, the simple GNG task, and the emotional GNG task. Bonferroni correction was applied on the ANOVA and post hoc analyses to compensate for a type I error. In addition, the correlation coefficient (*r*) between functional connectivity of each connection (Fisher’s *z* value described above) and subjects’ performance (omission error, commission error, and reaction time) was calculated across subjects. Similar to the statistical test, the correlation was significant when the *p*-value was less than or equal to 0.05.

## 3. Results

### 3.1. Performance

The subject’s performance, including omission errors, commission errors, and response time, during the simple and emotional GNG tasks was compared between sexes (Table 1). No statistically significant difference in performance was observed between sexes across any of these metrics (Table 1). Within sex comparisons showed that females committed significantly higher commission errors than omission errors in the simple GNG task (*t* = 4.2, *p*-value = 0.00013), but in the emotional GNG task, the omission errors were significantly greater than the commission errors (*p*-value = 0.04). Both sexes performed significantly better in the simple GNG task than in the emotional GNG task (fewer errors and faster response time).

### 3.2. Functional Connectivity in Three Conditions

Figure 2 shows the prefrontal connectivity during a vanilla baseline, a simple GNG task, and an emotional GNG task. Strong connectivity (*z*-value > 0.5, red edge in Figure 2) was observed in 13 right prefrontal connections, one inter-hemispheric connection, and four left medial prefrontal connections in the baseline, 10 right prefrontal connections, one inter-hemispheric connection, and two left medial prefrontal connections in the simple GNG, and 13 right prefrontal connections, two inter-hemispheric connection, and two left medial prefrontal connections in the emotional GNG. All connections with strong connectivity in the simple GNG overlapped with the ones in the baseline. Similarly, 16 out of 17 strong connections during the emotional GNG overlapped with those seen during the baseline.

The repeated measures ANOVA (120 tests for 120 connections, Bonferroni corrected critical *p*-value = 0.0004) comparing connectivity strength in all connections revealed no statistical difference between the three conditions for the whole group, males, and females (all *p*-values > 0.0004). In addition, within condition student *t*-test (360 tests for 120 connections and three conditions, Bonferroni corrected critical *p*-value = 0.0001) resulted in no significant difference in connectivity strength between sexes in all three conditions (all *p*-values > 0.0001).

### 3.3. Correlation between Functional Connectivity and Performance

Correlations between the omission errors, commission errors, response time, and functional connectivity during the tasks were calculated to examine the relationship between subject’s performance and brain connectivity. Figure 3 and Figure 4 display brain maps of the correlations between connectivity and subject’s performance during the tasks. In general, a negative correlation coefficient implies that fewer errors/shorter response time corresponds to higher connectivity (better performance → greater connectivity), while a positive correlation indicates the opposite (better performance → smaller connectivity).

During the simple GNG task, all connections with significant correlation (thick edges, Figure 3a,b,d,e) show a negative relationship between connectivity–omission error and connectivity–commission error in both sexes. The connectivity–response time correlation in the female group is negative in all except one significantly correlated connection (thick edges, Figure 3c). This means that greater simple GNG connectivity is associated with better task performance across both sexes.

The negative relationships in connectivity–omission error and connectivity–commission error are maintained in the male group during the emotional GNG (Figure 4d,e). A negative connectivity–omission error correlation is observed in all except one connection, and a negative connectivity–commission error correlation is observed in all connections with significant correlation. In contrast, a positive relationship in connectivity–omission error and connectivity–commission error appears in the female group. A positive connectivity–omission error correlation is presented in all significantly correlated connections in the female group (Figure 4a). In general, when considering the omission and commission errors, greater emotional GNG connectivity corresponds to a better task performance in the male group but a poorer task performance in the female group.

### 3.4. Correlation between the Baseline Connectivity with the Simple and Emotional Connectivity

All connections in the male group and all except two connections in the female group have a positive correlation coefficient between the baseline connectivity and the task connectivity (data not shown).

Figure 5 displays the connections that have low correlation coefficients (*p*-value > 0.05) between the baseline and the task connectivity, which are considered as task-specific connections. The female group recruited 11 connections during the simple GNG and nine connections during the emotional GNG, among which seven connections were common in both tasks (Figure 5a). The male group required four connections to perform the simple GNG and eight connections to perform the emotional GNG, among which one connection was common in both tasks (Figure 5b). All task-specific connections are either right hemisphere or inter-hemispheric connections.

## 4. Discussion

A high connectivity in connections within the right hemisphere, inter-hemisphere, and left medial prefrontal cortex found during both GNG tasks is in agreement with previous research [28], which emphasized the critical role of the prefrontal cortex in the motor response inhibition. In addition, in line with the studies of Duann et al. [18] and Davidow et al. [19], we found an association between a greater connectivity in the prefrontal cortexes and a better simple GNG performance. The finding of a connectivity–performance relation switch from a simple GNG to an emotional GNG in the female group but not in the male group suggests that the prefrontal functional connectivity of the male and female groups may have responded differently to a combination of emotional and inhibitory control. In general, since the strength of the cerebral functional connectivity depends on both a subject’s performance and sex, it is critical to consider these factors when comparing connectivity in different groups.

Based on the commonly used method, we found no connections that are specific for a GNG task, in that there was no significant difference in any connections between the baseline and task connectivity (ANOVA tests, Section 3.2). As aforementioned, the traditional method is not appropriate to compare resting and task connectivity in the prefrontal cortex, since this brain region expresses a high connectivity level in both conditions (Section 3.2, Figure 2). This study suggested a new method to explore GNG task-specific connections. With this approach, we have identified 13 connections in the female groups and 11 connections in the male groups that are specific to a GNG task. Interestingly, the female group prefrontal network recruited more connections (11 connections) during a simple GNG task than the male group (four connections), which implies that females may require more brain resources to perform a simple GNG than males. This finding is in line with the result found in Melynyte et al.’s study where they reported that females required more neural resources for a Go execution [29].

Most fMRI studies evaluating the brain circuits and areas involved in the response inhibition process have revealed multiple right prefrontal areas that are associated with this process including the right inferior frontal gyrus and right middle frontal gyrus [6,7,8]. Similar to these findings, most connections (nine connections in the female group and eight connections in the male group), which are found to be specific to the GNG task in the current study, are in the right hemisphere. The difference between the results from this study and previous fMRI studies is the involvement of the inter-hemispheric connections in the GNG task. We found four inter-hemispheric connections in the female group and three inter-hemispheric connections in the male group specific for the task. Our findings suggest that the left prefrontal cortex may be indirectly associated with the GNG task through its interaction with the right prefrontal cortex.

A limitation of this study lies in the use of the NIRS probe, which only covers the prefrontal cortex region. Currently, we can only investigate the prefrontal functional connectivity, but not inter-brain region connectivity (e.g., frontal-motor or frontal-sensory connectivity) during the GNG task. As the behavioral inhibition task may involve the pre-supplementary motor area [7], and the emotional task may activate additional neural regions, future study should cover other brain regions to examine the interaction between brain regions during a behavior inhibition and emotional regulation task.

## 5. Conclusions

This study investigated a sex-based functional connectivity in the prefrontal cortex during a simple and emotional Go/No-Go task, which was then correlated to the subject’s performance. We found a strong connectivity in the right hemisphere, inter-hemisphere, and left medial prefrontal cortex in all conditions. No differences in the Go/No-Go performance nor the prefrontal connectivity were found between the male and female groups. Both sex groups had a positive correlation between the prefrontal connectivity and the simple GNG performance. However, although the male group had a positive correlation, the female group expressed a negative correlation between the prefrontal connectivity and the emotional GNG performance. Additionally, this study found that females recruited a greater number of brain connections to perform a behavior inhibition task than males.

## Figures and Tables

**Figure 1 brainsci-11-00909-f001:**
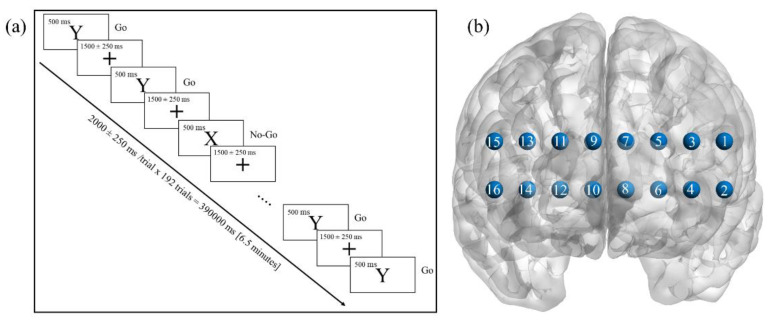
Experimental protocol; (**a**) experimental paradigm during a simple GNG task, letter Y: Go stimulus, letter X: No-Go stimulus, + sign: inter-stimulus rest. Letter Y was replaced by a photo of a neutral face and letter X was replaced by a photo of a happy or angry face during the emotional GNG task; (**b**) location of 16 fNIRS channels in a brain model.

**Figure 2 brainsci-11-00909-f002:**
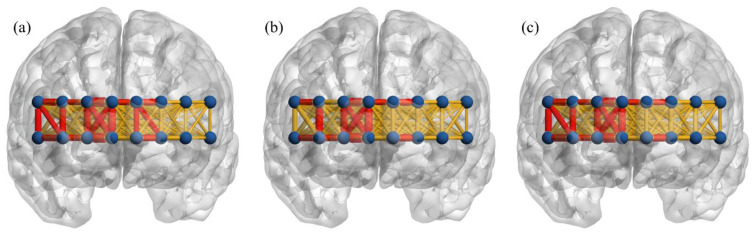
Averaged functional connectivity across all subjects in three conditions; (**a**) vanilla baseline, (**b**) simple GNG, and (**c**) emotional GNG. Red edge: connections that have connectivity greater than 0.5, yellow edge: other connections.

**Figure 3 brainsci-11-00909-f003:**
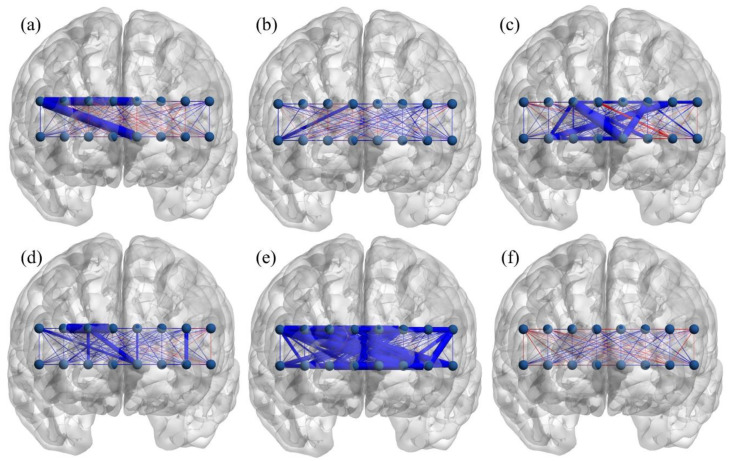
Brain map depicting correlation coefficients between the simple GNG connectivity and subject performance; (**a**–**c**) female, (**d**–**f**) male; (**a**,**d**), omission error, (**b**,**e**) commission error, (**c**,**f**) response time. Red: positive correlation, blue: negative correlation, thick edge: connection with significant correlation coefficient (*p*-value < 0.05).

**Figure 4 brainsci-11-00909-f004:**
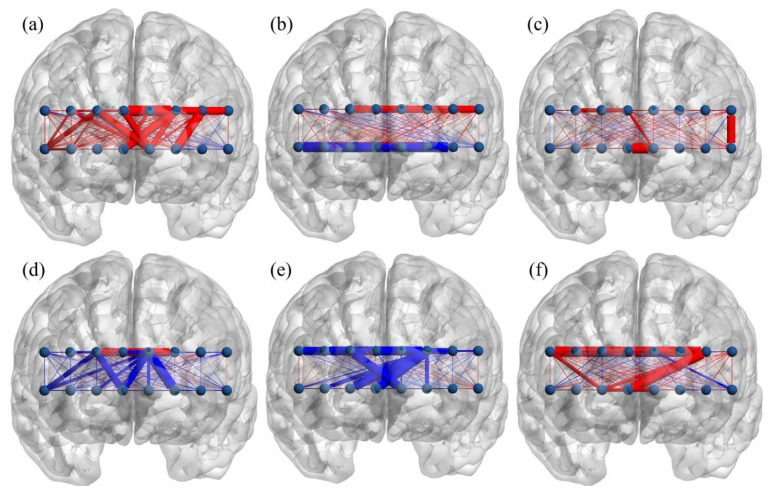
Brain map depicting correlation coefficient between the emotional GNG connectivity and subject performance; (**a**–**c**) female, (**d**–**f**) male; (**a**,**d**) omission error, (**b**,**e**) commission error, (**c**,**f**) response time. Red: positive correlation, blue: negative correlation, thick edge: connection with significant correlation coefficient (*p*-value < 0.05).

**Figure 5 brainsci-11-00909-f005:**
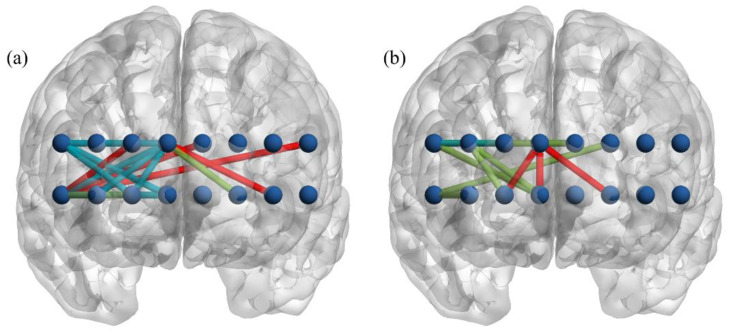
Connection with low correlation coefficients between conditions; (**a**) female, (**b**) male. Red: baseline–simple GNG, green: baseline–emotional GNG, cyan: both baseline–simple GNG and baseline–emotional GNG.

**Table 1 brainsci-11-00909-t001:** Subject’s performance: mean and standard deviation of omission errors, commission errors, and response time during the simple and emotional GNG task by sex.

GNG	Omission Errors				Commission Errors				Response Time (ms)			
	Female	Male	*t*	*p*-Value	Female	Male	*t*	*p*-Value	Female	Male	*t*	*p*-Value
Simple	0.7 (1.5)	2.5 (5.7)	1.32	0.19	4.9 (4.0)	4.8 (3.6)	0.08	0.93	436 (51)	453 (84)	0.75	0.46
Emotional	28.4 (40.0)	22.2 (25.4)	0.59	0.56	12.8 (11.3)	11.9 (12.4)	0.24	0.81	703 (128)	745 (179)	0.83	0.41

## Data Availability

The data presented in this study are available on request from the corresponding author. The data are not publicly available due to privacy restriction.

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
