# Peer review of "Comparison of Functional Connectivity in the Prefrontal Cortex during a Simple and an Emotional Go/No-Go Task in Female versus Male Groups: An fNIRS Study"

_brainsci, 2021, doi:10.3390/brainsci11070909_

Round 1

Reviewer 1 Report

Dear Prof. Zou,   Here below are the comments for this paper:   The present study investigated prefrontal functional connectivity during a behavioral inhibition task and its correlation with subject's performance; and identified connections that are specific to the Go/No-Go task. And minor revision is needed.   1. Introduction: [1] line 43: Does the "typical population" here refer to the male and the female? If yes, please clarify the reason and necessity. If not, please give explanation and the reason to do that.   [2] line 70 & 71: please clarify "the neural circuitry" and "emotion processing areas; and provided relevant references.   [3] line 80-82: which kind of inhibition deficit can the GNG task differentiate? Is it the same thing as inhibition deficit of ADHD?   [4] line 82-84: Please give more details on estrogen variation and its influences on GNG performance among the females and males.   [5] line 92-93: The following expression seems incorrect. Please revise it and write in more academic way.   'When a task provokes activation in two different regions, a strong connectivity is observed between these regions.'   [6] line 96: Due to the disadvantage of fNIRS, the calculation of structural connectivity is impossible.   [7] line 98-99: As discribed in the methods part, only prefrontal lobe was covered with probes and monitored. Only regional connectivity is calculated in the present study, rather than the network-level dynamics.   [8] line 105: In the last paragraph of introduction, it seems that you selected the prefrontal lobe as targeted region because of its relationship with GNG task. As you also compared the simple and emotion GNG tasks, you need to associate the prefrontal lobe with emotion. The current reason for targeted region selection is not persuasive.   2. Methods [1] It would be better to add a figure for your GNG task. [2] line 191-192: please clarify your independent variable (HbO? HbR? β value?) [3] line 198-200: please clarify your within-/between-subject factors in the 3-way repeated measures ANOVA.

Reviewer 2 Report

Authors explored on 42 healthy adults prefrontal functional connectivity during a behavioral inhibition tasks (vanilla baseline, and a simple and emotional Go/No-Go task) during a 16-channel near infrared spectroscopy (fNIRS) registration. Functional connectivity was calculated from fNIRS signals and correlated to the Go/No-Go performance. Strong connectivity was found in both the tasks in the right hemisphere, interhemispherically, and left medial prefrontal cortex. Notably, females showed a lower emotional Go/No-Go connectivity while performing better on the task.

This is a well-designed and interesting study. In my opinion the paper is of potential interest for publication on Brain Sciences. However, I believe that several sections of the manuscript need to be improved. In my view the manuscript is not ready for publication yet.

These are my comments:

  1.  In introduction authors neglected a relevant metanalysis:

“Reviews of fMRI studies investigating behavioral inhibition using the GNG task show that frontal areas such as the pre-supplementary motor area, insula, and medial prefrontal cortex are activated during these tasks [5,6] …  A meta-analysis further investigating how the varied parameters of GNG tasks affect activation found that alterations in the ‘Go’ to ‘No-Go’ trial ratio, the number of ‘No-Go’ stimuli and working memory load (e.g., more complex stimuli, more than one ‘Go’ or ‘No-Go’ target) changed which areas were activated [7]. Specifically, the authors note that certain areas, such as the pre-supplementary motor area, may not play a direct role in inhibition because activation in this area appeared to be attributed to increased working memory load [7].”

Authors must improve this section adding more recent literature on cognitive control. In particular, Gavazzi et al., (2020) should be at least mentioned, because these authors explored recently with ALE metanalyses the domain of cognitive control, re-runned Simmonds et al., (2008) dataset (that authors cite several times in the paper) with the updated ALE algorithm  and proposed a model of cognitive control.

Gavazzi, G.; Giovannelli, F.; Currò, T.; Mascalchi, M. Contiguity of proactive and reactive inhibitory brain areas: A cognitive model based on ALE meta-analyses. Brain Imaging Behav. 2020.

  1. Please describe more in details the task section because the tasks were not well explained. I suggest an additional figure (or a supplementary figure - if possible).

  1. In methods. Please indicate more precisely what tests are considered when applying Bonferroni corrections because it is not clear. Did the authors applied the Bonferroni corrections to all analysis or not? If not where they have been applied and where not.

  1. In discussion. I believe that the authors should employ more recent model of cognitive control to interpret their results. In particular, Gavazzi et al. (2020) proposed a new model of cognitive control based on ALE metanalyses (the proactive-reactive model of cognitive control or P-R M). This can improve the interpretation of the results presented in this work.

Gavazzi, G.; Giovannelli, F.; Currò, T.; Mascalchi, M. Contiguity of proactive and reactive inhibitory brain areas: A cognitive model based on ALE meta-analyses. Brain Imaging Behav. 2020.

  1. I think that authors should discuss differences and similarities betweent he present results (obtained with fNIRS) with what is typically observed with fMRI.

Round 2

Reviewer 2 Report

The manuscript is ready to be published